# Performance of formal smell testing and symptom screening for identifying SARS-CoV-2 infection

James W. Keck[1]*, Matthew Bush[2], Robert Razick[3], Setareh Mohammadie[3], Joshua Musalia[4], Joel Hamm[3]

1 Department of Family and Community Medicine, University of Kentucky, Lexington, KY, United States of America, 2 Department of Otolaryngology–Head and Neck Surgery, University of Kentucky, Lexington, KY, United States of America, 3 Department of Emergency Medicine, University of Kentucky, Lexington, KY, United States of America, 4 College of Medicine, University of Kentucky, Lexington, KY, United States of America

* James.Keck@uky.edu

## Abstract

### Background

Altered sense of smell is a commonly reported COVID-19 symptom. The performance of smell testing to identify SARS-CoV-2 infection status is unknown. We measured the ability of formal smell testing to identify SARS-CoV-2 infection and compared its performance with symptom screening.

### Methods

A convenience sample of emergency department patients with COVID-19 symptom screening participated in smell testing using an eight odor Pocket Smell Test (PST). Participants received a SARS-CoV-2 viral PCR test after smell testing and completed a health conditions survey. Descriptive analysis and receiver operating characteristic (ROC) curve models compared the accuracy of smell testing versus symptom screening in identifying SARS-CoV-2 infection.

### Results

Two hundred and ninety-five patients completed smell testing and 87 (29.5%) had a positive SARS-CoV-2 PCR test. Twenty-eight of the SARS-CoV-2 positive patients (32.2%) and 49 of the SARS-CoV-2 negative patients (23.6%) reported at least one of seven screening symptoms (OR = 1.54, P = 0.13). SARS-CoV-2 positive patients were more likely to have hyposmia (≤5 correctly identified odors) than SARS-CoV-2 negative patients (56.1% vs. 19.3%, OR = 5.36, P<0.001). Hyposmia was 52.9% (95% CI 41.9%-63.7%) sensitive and 82.7% (95% CI 76.9%-87.6%) specific for SARS-CoV-2 infection. Presence of ≥1 screening symptom was 32.2% (95% CI 22.6%-43.1%) sensitive and 76.4% (70.1%-82.0%) specific for SARS-CoV-2 infection. The ROC curve for smell testing had an area under the curve (AUC) of 0.74 (95% CI 0.67–0.80). The ROC curve for symptom screening had lower

**Data Availability Statement:** All relevant data are within the paper and its Supporting Information files.

**Funding:** This work supported by the National Institutes of Health (NIH) National Center for Advancing Translational Sciences [grant number UL1TR001998] received by authors JWK, MB, and JH. The content is solely the responsibility of the authors and does not necessarily represent the official views of the NIH. The funders had no role in study design, data collection and analysis, decision to publish, or preparation of the manuscript.

**Competing interests:** The authors have declared that no competing interests exist.

discriminatory accuracy for SARS-CoV-2 infection (AUC = 0.55, 95% CI 0.49–0.61, P<0.001) than the smell testing ROC curve.

## Conclusion

Smell testing was superior to symptom screening for identifying SARS-CoV-2 infection in our study.

## Introduction

Mitigating community transmission of SARS-CoV-2 has proven challenging, in part because of the high proportion of infected individuals who are asymptomatic or presymptomatic. Asymptomatic and presymptomatic individuals may cause 40% of new SARS-CoV-2 infections [1, 2], and asymptomatic SARS-CoV-2 infection occurs about 50% of the time [3]. These cases evade the ubiquitous symptom-based screening strategies used by employers, schools, and businesses. Theoretically, symptom-based screening for SARS-CoV-2 infection is only as sensitive as the percentage of infected cases with symptoms. A more sensitive screening tool that also identifies asymptomatic cases could focus clinical testing and quarantine activities and better mitigate community transmission.

A frequently reported symptom by people with confirmed SARS-CoV-2 infection is altered sense of smell or taste. In a cohort of 2.6 million people in the United Kingdom, 65% of participants with a positive SARS-CoV-2 viral test reported a subjective loss of smell making it the most frequently reported symptom [4]. However, people are often unaware that their sense of smell is diminished [5]. Smell testing, an objective method of measuring olfaction, is more likely to identify diminished olfaction than self-report [6]. Approximately 20% of the general population has olfactory dysfunction on objective testing [7]. The discrepancy in self-reported smell alteration compared to measured smell alteration was seen in a study of patients hospitalized with COVID-19 in Iran, where 59 of 60 patients had altered olfaction on smell testing, but only 21 of the 60 reported alteration in smell or taste function [8].

The comparative accuracy of smell testing versus symptom screening to identify whether someone is infected with SARS-CoV-2 is unknown. We hypothesized that smell testing using "scratch and sniff" odor cards is superior to symptom screening in identifying SARS-CoV-2 infection. To test this hypothesis, we prospectively conducted smell testing and symptom screening of ambulatory patients who then received a SARS-CoV-2 viral PCR test to measure the performance of smell testing and symptom screening for identifying SARS-CoV-2 infection.

## Methods

### Study population

We prospectively enrolled adult patients who sought care at the emergency department of an academic medical center. Our convenience sample included two groups of patients with anticipated SARS-CoV-2 testing: 1) patients with reported COVID-19 exposure or a positive symptom screen at triage (≥1 of the following self-reported symptoms chosen for their prevalence in COVID-19 illness [9]: fever, shortness of breath, cough, chills, sore throat, loss of taste and/or smell, or body aches) or 2) planned hospital admission (all admitted patients tested for SARS-CoV-2 regardless of admitting diagnosis for infection prevention). We excluded

patients with an altered level of consciousness and minors. Study enrollment via convenience sampling occurred throughout the week by multiple clinicians and researchers. Participants provided written informed consent prior to study data collection. The study protocol was reviewed and approved by the University of Kentucky Institutional Review Board (protocol #61519).

## Sample size

We used Buderer's formula [10] to estimate the sample size with $\alpha = 0.05$, a marginal error of 0.1 and the hypothesis that 85% of people with SARS-CoV-2 infection have measurable loss of smell. This hypothesis was based on self-reported loss of smell in 65% of those with SARS-CoV-2 infection [4] and measured alteration in smell in 98% of hospitalized COVID-19 patients [8]. At the time of study design, SARS-CoV-2 test positivity was 5% which yielded a sample size of 980 patients. During study recruitment local SARS-CoV-2 test positivity increased to about 10%, and an interim analysis of our convenience sample showed SARS-CoV-2 positivity of 25%. This increase in disease prevalence reduced the estimated sample size to 196 patients, and we ended study recruitment with 308 patients.

## Study procedures and data collection

Prior to study enrollment patients completed hospital protocol-driven SARS-CoV-2 symptom screening. With informed consent, a member of the study team provided a brief survey and a self-administered smell test. The survey collected data on patient demographics, health conditions potentially affecting sense of smell, and self-reported problems with smell. We used the validated National Health and Nutrition Examination Survey (NHANES) pocket smell test (PST) (Sensonics, Inc., Haddon Heights, NJ) a self-administered "scratch and sniff" smell test [11]. Each participant completed versions A and B of the PST, and each version had four distinct odors (listed in Fig 1) to identify from a multiple choice list of smells. SARS-CoV-2 viral testing occurred after smell testing as part of routine clinical care. Study clinicians were blinded to the results of the smell tests. Hospital staff obtained a nasopharyngeal swab for SARS-CoV-2 RT-PCR testing with either the Abbott Alinity m2000 (Abbott Laboratories,

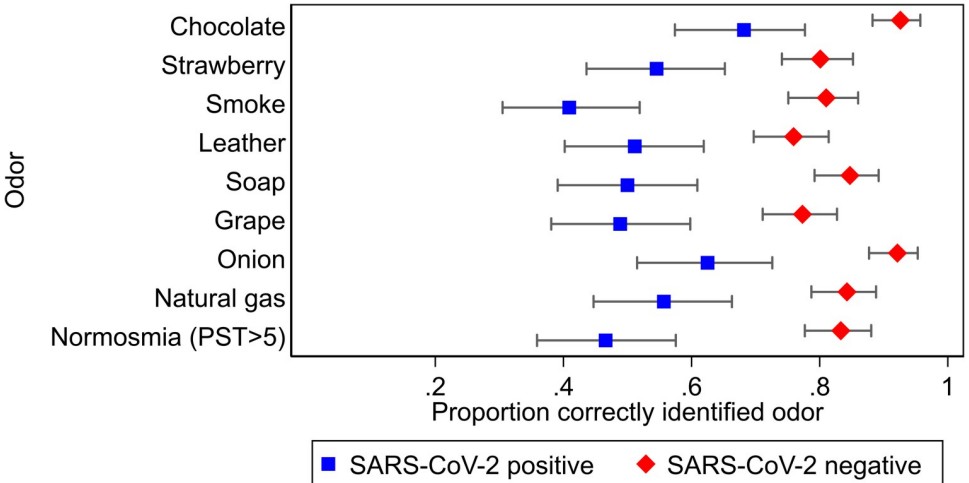

**Fig 1. Smell testing odor discrimination by SARS-CoV-2 infection status.** Blue squares represent the proportion of SARS-CoV-2 positive patients that correctly identified the odor; red diamonds the proportion of SARS-CoV-2 negative patients that identified the odor. 95% confidence intervals for the point estimates are shown with whiskers.

Santa Clara, CA) or BD Max (Beckton Dickinson, Franklin Lakes, NJ) platform. RT-PCR testing happened for all patients with a positive symptom screen, close contact with a known COVID-19 case, or planned hospital admission per hospital protocol. Study and clinical data were entered into REDCap, a secure, cloud-based data storage platform.

## Statistical analysis

We excluded nine study encounters (2.9%) because the patient had previously participated in the smell testing study. We additionally removed four of the remaining 299 participants (1.3%) from the analysis due to incomplete smell testing data. Data for the other variables was complete for the remaining 295 participants. We described our patient sample using means, proportions, and standard deviations. We used chi-squared tests to assess the distribution of patients across categorical variables, t tests for normally distributed continuous variables, Wilcoxon rank-sum test for non-normally distributed variables and used logistic regression and odds ratios to describe associations with our primary outcome, SARS-CoV-2 status, and the classifier variables of odor discrimination and screening symptoms. We classified patients who correctly identified six or more of the eight odors as normosmic and those who correctly identified fewer than 6 odors as hyposmic per the NHANES classification system [5]. We calculated the sensitivity, specificity, and predictive values of smell testing, self-reported symptoms, and measured fever in identifying patients with SARS-CoV-2 infection as determined by viral RT-PCR testing. We used receiver operating characteristic (ROC) curves and calculated the area under the curve (AUC) to compare the performance of six screening approaches for predicting SARS-CoV-2 infection. The models used the following classifiers: 1) number of the eight PST odors accurately identified; 2) presence of hyposmia defined as correctly identifying five or fewer odors correctly on the PST; 3) number of self-reported screening symptoms (out of seven); 4) presence of measured fever (body temperature $\geq 100.4°F$); 5) two-odor (smoke and soap) performance on the PST; and 6) adjusted model that used the eight PST odors adjusted by the covariates age, gender, corticosteroid nasal spray use, measured fever, and cough. We developed the two-odor model by calculating the performance (AUC) of each of the eight PST odors to identify SARS-CoV-2 infection and using the two odors with the best performance (largest AUC). For the adjusted model age and gender were selected a priori and the other covariates were included because they were statistically associated with smell testing performance (corticosteroid nasal spray use) or with SARS-CoV-2 infection (fever and cough). We conducted a ROC subgroup analysis that excluded patients with reported previous positive SARS-CoV-2 test or COVID-19 exposure and a second subgroup analysis of asymptomatic patients to assess the predictive performance of smell testing in asymptomatic patients. To compare the predictive performance of ROC models we used a chi-squared statistic to test for differences in AUC between models. Reported P-values are two-sided, and we considered P<0.05 statistically significant. Author JWK conducted the analyses with STATA version 15.1 (StataCorp, College Station, TX).

## Results

From October 16, 2020, to February 15, 2021, 295 unique patients completed smell testing, and 87 (29.5%) of those patients tested positive for SARS-CoV-2 by RT-PCR. Study participants were 52.2% (154/295) female, 83.3% (246/295) white, and had a mean age of 45.6 years (SD = 17.8; IQR 30–59). There were no statistically significant demographic differences between the participants that tested positive for SARS-CoV-2 and the participants that tested negative (Table 1). Seventy-seven (26.1%) participants reported at least one symptom, which included 49 (23.6%) SARS-CoV-2 negative patients and 28 (32.2%) SARS-CoV-2 positive

**Table 1. Patient demographics and symptoms by SARS-CoV-2 infection status.**

| | SARS-CoV-2 Negative (N = 208) | | SARS-CoV-2 Positive (N = 87) | | |
|---|---|---|---|---|---|
| **Characteristic** | **n** | **%** | **n** | **%** | **P** |
| Demographic | | | | | |
| Age (years; mean, SD) | 44.5 | 17.2 | 48.1 | 19.1 | 0.16 |
| Female | 111 | 53.4% | 43 | 48.3% | 0.43 |
| Race | | | | | 0.11 |
| Asian | 0 | 0.0% | 2 | 2.3% | |
| Black | 28 | 13.5% | 17 | 19.5% | |
| White | 178 | 85.6% | 68 | 77.0% | |
| Multiracial | 1 | 0.5% | 0 | 0.0% | |
| Hispanic | 7 | 3.4% | 3 | 3.5% | 0.77 |
| Symptom/exposure screening | | | | | |
| Previous positive SARS-CoV-2 test or exposure to someone with COVID-19 | 8 | 3.9% | 20 | 23.0% | <0.001 |
| Reported fever | 16 | 7.7% | 10 | 11.5% | 0.29 |
| Shortness of breath | 25 | 12.0% | 17 | 19.5% | 0.09 |
| Cough | 22 | 10.6% | 22 | 25.3% | 0.001 |
| Chills | 16 | 7.7% | 11 | 12.6% | 0.18 |
| Sore throat | 13 | 6.3% | 9 | 10.3% | 0.22 |
| Loss of taste and/or smell | 5 | 2.4% | 9 | 10.3% | 0.003 |
| Body aches | 22 | 10.6% | 12 | 13.8% | 0.43 |
| Any reported symptom | 49 | 23.6% | 28 | 32.2% | 0.13 |
| Maximum recorded temperature in ED (mean, SD in ˚F) | 98.5 | 0.94 | 98.8 | 1.2 | 0.01 |
| Recorded fever in ED (T > = 100.4˚F (38˚C)) | 8 | 3.9% | 7 | 8.1% | 0.13 |

ED = emergency department; SD = standard deviation

patients (OR = 1.54; P = 0.13). Cough and loss of taste and/or smell were the only symptoms significantly more common in SARS-CoV-2 positive patients (Table 1).

Most patients (n = 213; 72.2%) had normal smell function (PST score of 6 to 8) on smell testing. Hyposmia was more common in older patients (OR 1.17 per 10 years of age, 95% confidence interval (CI) 1.01–1.35, P = 0.03), patients who reported a history of loss of smell (OR 3.44, 95% CI 1.56–7.61, P = 0.002), and in patients who used nasal sprays (OR 2.12, 95% CI 1.16–3.91, P = 0.02) (S1 Table). We did not observe associations between hyposmia and health conditions that can affect the sense of smell (S1 Table).

Of the 87 patients testing positive for SARS-CoV-2, 71 (81.6%) misidentified at least one odor in the PST. Individual odor identification accuracy by patient SARS-CoV-2 status appears in Fig 1. SARS-CoV-2 positive patients were more likely to have hyposmia (misidentify at least three of the eight odors) as compared to SARS-CoV-2 negative patients (52.3% vs 17.3%; OR = 5.36; 95% CI 3.08–9.43; P<0.001). Hyposmia was 52.9% (95% CI 41.9%-63.7%) sensitive and 82.7% (95% CI 76.9%-87.6%) specific for SARS-CoV-2 infection. Symptom screening (≥1 symptom) was 32.2% sensitive (95% CI 22.6%-43.1%) and 76.4% specific (95% CI 70.1%-82.0%) for SARS-CoV-2 infection, and measured fever had 8.0% sensitivity (95% CI 3.3%-15.9%) and 96.2% specificity (95% CI 92.6%-98.4%) for SARS-CoV-2 infection. The positive predictive values (PPV) and negative predictive values (NPV) for SARS-CoV-2 infection with smell testing, symptom screening, and measured temperature are shown in Table 2 under the scenarios of 1%, 5%, and 10% prevalence of infection.

**Table 2. Sensitivity, specificity, and predictive values of smell testing, symptom screening, and body temperature measurement for identifying SARS-CoV-2 infection.**

| | Smell testing (hyposmia: PST≤5) | | | Symptom screening (≥1 symptom) | | | Measured temperature (temp≥100.4˚F) | | |
|---|---|---|---|---|---|---|---|---|---|
| | SARS-CoV-2 Prevalence | | | SARS-CoV-2 Prevalence | | | SARS-CoV-2 Prevalence | | |
| | 1.0% | 5.0% | 10.0% | 1.0% | 5.0% | 10.0% | 1.0% | 5.0% | 10.0% |
| PPV | 3.0% | 13.9% | 25.3% | 1.4% | 6.7% | 13.2% | 2.1% | 9.9% | 18.9% |
| NPV | 99.4% | 97.1% | 94.0% | 99.1% | 95.5% | 91.0% | 99.0% | 95.2% | 90.4% |
| False positive rate | 17.3% | | | 23.6% | | | 3.8% | | |
| False positives (per 1,000 screened) | 171 | 164 | 156 | 233 | 224 | 212 | 38 | 37 | 35 |
| False negative rate | 47.1% | | | 67.8% | | | 92.0% | | |
| False negatives (per 1,000 screened) | 5 | 24 | 47 | 7 | 34 | 68 | 9 | 46 | 92 |
| Correctly classified | 824 | 812 | 797 | 760 | 742 | 720 | 953 | 917 | 873 |

PST = pocket smell test; PPV = positive predictive value; NPV = negative predictive value

The receiver operating characteristic (ROC) curve for smell testing to classify SARS-CoV-2 infection yielded an area under the curve (AUC) of 0.74 (95% CI 0.67–0.80: Table 3 and Fig 2A). The ROC curve using reported symptoms to classify SARS-CoV-2 infection had an AUC of 0.55 (95% CI 0.49–0.61: Fig 2B). The measured fever ROC curve had an AUC of 0.52 (95% CI 0.49–0.55: Fig 2C).

We evaluated two additional ROC models that used smell testing as the classifier. The hyposmia (PST score≤5) model had an AUC of 0.68 (95% CI: 0.62–0.74), and a 2-odor model using the odors smoke and soap (largest independent AUCs for predicting SARS-CoV-2 infection; S1 Fig) had an AUC of 0.75 (95% CI 0.69–0.80: Fig 2D). The sensitivity, specificity, and predictive values of the 2-odor model were similar to the 8-odor model (data not shown). A multi-classifier ROC model adding classifiers significantly associated with SARS-CoV-2 infection (corticosteroid nasal spray use, cough, measured fever) plus age and gender to the 8-odor model marginally increased the AUC to 0.79 (95% CI 0.73–0.85). ROC smell testing subgroup analyses that excluded patients with reported symptoms (AUC = 0.76; 95% CI 0.69–0.84) and patients with reported COVID-19 exposure or previous positive SARS-CoV-2 test (AUC = 0.75; 95% CI 0.68–0.81) performed similarly to the 8-odor ROC model using the full data set.

**Table 3. Receiver Operating Characteristic (ROC) models and subgroup analyses for predicting SARS-CoV-2 infection.**

| Model | Variable(s) | Observations | AUC | 95% CI | | P* |
|---|---|---|---|---|---|---|
| 8-odor | 8 odors | 295 | 0.74 | 0.67 | 0.80 | |
| Hyposmia | Hyposmia (PST≤5) | 295 | 0.68 | 0.62 | 0.74 | 0.003 |
| Symptoms | 7 symptoms | 295 | 0.55 | 0.49 | 0.61 | <0.001 |
| Fever | Body temperature ≥100.4˚F | 295 | 0.52 | 0.49 | 0.55 | <0.001 |
| 2-odor | Smoke + soap odors | 295 | 0.75 | 0.69 | 0.80 | 0.60 |
| Adjusted | 8 odors, age, gender, corticosteroid nasal spray use, measured fever, cough | 295 | 0.79 | 0.73 | 0.85 | 0.02 |
| **Subgroup analyses** | | | | | | |
| No symptoms | 8 odors | 218 | 0.76 | 0.69 | 0.84 | |
| No COVID test/exposure | 8 odors | 267 | 0.75 | 0.68 | 0.81 | |

AUC = area under the curve; CI = confidence interval; PST = pocket smell test

*As compared to the 8-odor model

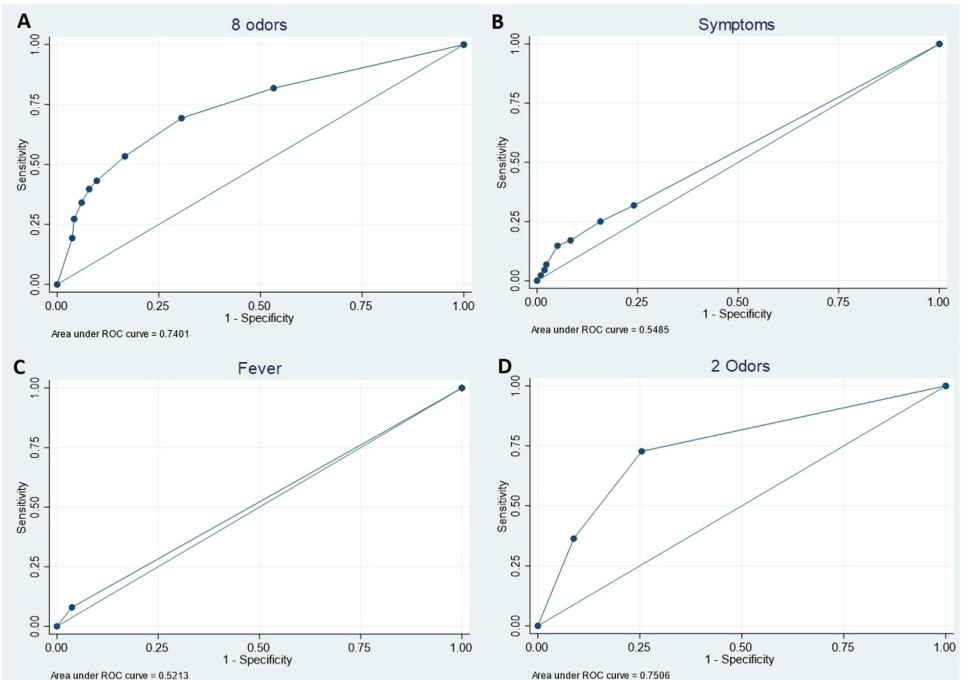

**Fig 2.** Receiver Operating Characteristic (ROC) curves to identify SARS-CoV-2 infection: a) Smell testing with 8 odors; b) Symptom screening with 7 self-reported symptoms; c) Temperature screening for fever ($\geq$100.4°F); d) Smell testing with 2 odors (smoke and soap).

## Discussion

Smell testing identified SARS-CoV-2 infection with greater sensitivity and specificity than COVID-19 symptom screening in our population. The prospective design of our study, which used objective measurements of smell paired with SARS-CoV-2 PCR testing yielded robust estimates of the performance of smell testing in symptomatic and asymptomatic patients. Our study is unique in that it prospectively assessed patient sense of smell prior to ascertaining SARS-CoV-2 status; previous work in this area has tested olfaction in known COVID-19 patients [8, 12, 13] or assessed self-reported alterations in smell in patients with known SARS-CoV-2 infection [4, 14–16], with neither approach supporting an accurate assessment of smell testing as a screening tool for SARS-CoV-2 infection in symptomatic and asymptomatic patients.

An optimal SARS-CoV-2 screening test should perform well regardless of health status and preexisting health conditions. Preexisting health conditions, like allergic rhinitis and chronic sinusitis, that may affect olfaction did not affect patient smell testing performance (S1 Table). The consistent performance of smell testing in our pragmatic cohort suggests that implementation of this screening tool does not need individual-level information (e.g., history of allergic rhinitis or age) to adjust smell testing results nor are more complex predictive models with multiple variables needed.

An ideal SARS-CoV-2 smell screening tool would be of low cost and require minimal time to administer. The 8-odor PST took less than 2 minutes to complete for most participants. Our study suggests that we can further streamline smell testing without affecting the performance of the screening test, as a simplified smell testing model based on the two odors with the largest individual AUCs (smoke and soap) performed similarly to the eight-odor model in identifying

SARS-CoV-2 infection. A self-administered two-odor smell test is inexpensive and efficient, making it feasible to screen many people quickly.

Smell testing more accurately identified SARS-CoV-2 infection in the subgroup of asymptomatic patients as compared to the entire sample of symptomatic and asymptomatic patients, although this was not statistically significant. The performance of smell testing in asymptomatic patients suggests its utility as a SARS-CoV-2 screening tool in asymptomatic populations, like employees at congregate work settings. Larremore et al. modeled the effectiveness of smell testing to limit SARS-CoV-2 transmission and found every third day smell testing more effective than weekly RT-PCR testing when smell testing sensitivity was 75%, which is similar to the sensitivity of the two-odor model [17]. Interestingly, formal smell testing in our study was more sensitive than antigen testing in identifying SARS-CoV-2 infection in asymptomatic people. According to a large Cochrane meta-analysis, antigen testing was 58% sensitive and over 99% specific for SARS-CoV-2 when compared to RT-PCR in people without symptoms [18].

Symptom screening to identify SARS-CoV-2 infection was barely better than the flip of a coin in our study, which agrees with Gerkin et al. who found non-olfactory, non-gustatory symptoms unhelpful in identifying SARS-CoV-2 infection [19]. Menni et al. used self-reported symptoms and SARS-CoV-2 test data collected via an app to develop a symptom-based algorithm with 65% sensitivity in identifying SARS-CoV-2 infection [4]. This large study of self-reported data is limited by selection bias (those who chose to enroll via the app are likely not representative of the general population) and measurement bias, in that only a small subset (0.64%) of self-selected participants were tested for SARS-CoV-2. The large untested fraction of participants had a substantially lower frequency of reported symptoms suggesting few asymptomatic participants received SARS-CoV-2 tests. A second, similar app-based study found that self-reported symptoms were 70% sensitive in identifying SARS-CoV-2 infection using Menni et al.'s symptom-based algorithm and had the same shortcomings, namely selection and measurement bias with only 1.1% of the participants reporting SARS-CoV-2 test results [16].

Our study has several limitations. First, we recruited a convenience sample of patients seeking care at the emergency department, and we cannot generalize the findings of our study to other populations, such as asymptomatic people in the general population. However, in a subgroup analysis that excluded patients with reported COVID-19 symptoms, smell testing was as good and potentially better at identifying SARS-CoV-2 infection compared to its performance in the entire sample. Second, some of our participants with positive SARS-CoV-2 PCR tests may have been previously infected with SARS-CoV-2 with prolonged viral shedding and recovery of normal olfaction, which would decrease the calculated sensitivity of smell screening for SARS-CoV-2 infection. A subgroup analysis that excluded patients with self-reported prior SARS-CoV-2 test and/or recent COVID-19 exposure demonstrated similar smell testing performance compared to the primary analysis. Third, the PST provides four multiple choice options for each odor, forcing the test subject to provide a response even when they are unable to determine the odor. This characteristic of the PST inflates PST scores leading to potential underascertainment of hyposmia and decreased sensitivity of smell testing for SARS-CoV-2. Fourth, the epidemiologic context of the pandemic during our study period (e.g., predominant circulating virus variants and vaccine coverage) may influence study results and impact their generalizability to other pandemic contexts.

## Conclusions

In conclusion, smell testing performed well in identifying SARS-CoV-2 infection, while symptom screening and measured temperature, which are widely used in many settings, did not

reliably discriminate SARS-CoV-2 infection in our population. A subset of two odors from the smell test (smoke and soap) performed similarly in identifying SARS-CoV-2 infection as the full eight-odor test and could be a simple, affordable screening tool. Additional studies on the performance of smell testing in asymptomatic populations, e.g., healthy workers, can validate the use of smell screening to risk stratify people for SARS-CoV-2 clinical testing.

## Supporting information

**S1 Table. Demographics, health history, symptoms, and SARS-CoV-2 status by smell testing performance.**
(DOCX)

**S1 Fig. Receiver Operating Characteristic (ROC) curves for predicting SARS-CoV-2 infection by individual odor.**
(DOCX)

**S1 Dataset.**
(CSV)

## Acknowledgments

We thank Dr. Beth Lacy, Dr. Karen Roper, and Dr. Zach Porterfield for their constructive review of the manuscript. We are grateful for the efforts of Theresa Mims and Ronda Petrey to recruit patients and collect study data.

## Author Contributions

**Conceptualization:** James W. Keck, Joel Hamm.

**Data curation:** Joshua Musalia.

**Formal analysis:** James W. Keck.

**Funding acquisition:** James W. Keck, Matthew Bush, Joel Hamm.

**Investigation:** Robert Razick, Setareh Mohammadie, Joel Hamm.

**Methodology:** James W. Keck, Matthew Bush, Joel Hamm.

**Project administration:** Joel Hamm.

**Supervision:** Joel Hamm.

**Writing – original draft:** James W. Keck, Setareh Mohammadie.

**Writing – review & editing:** Matthew Bush, Robert Razick, Joshua Musalia, Joel Hamm.

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
