## [Decision Letter · Decision Letter 0]

31 Dec 2021

PONE-D-21-38950Performance of formal smell testing and symptom screening for identifying SARS-CoV-2 infectionPLOS ONE

Dear Dr. Keck,

Thank you for submitting your manuscript to PLOS ONE. After careful consideration, we feel that it has merit but does not fully meet PLOS ONE’s publication criteria as it currently stands. Therefore, we invite you to submit a revised version of the manuscript that addresses the points raised during the review process.

We look forward to receiving your revised manuscript.

Kind regards,

Muhammad Tarek Abdel Ghafar, M.D

Academic Editor

PLOS ONE

Journal Requirements:

Reviewers' comments:

Reviewer's Responses to Questions

**Comments to the Author**

1. Is the manuscript technically sound, and do the data support the conclusions?

Reviewer #1: Yes

Reviewer #2: Yes

Reviewer #3: Yes

Reviewer #4: Yes

Reviewer #5: Yes

Reviewer #6: Yes

Reviewer #7: Yes

2. Has the statistical analysis been performed appropriately and rigorously? 

Reviewer #1: I Don't Know

Reviewer #2: Yes

Reviewer #3: Yes

Reviewer #4: Yes

Reviewer #5: Yes

Reviewer #6: Yes

Reviewer #7: Yes

3. Have the authors made all data underlying the findings in their manuscript fully available?

Reviewer #1: Yes

Reviewer #2: Yes

Reviewer #3: Yes

Reviewer #4: Yes

Reviewer #5: Yes

Reviewer #6: Yes

Reviewer #7: Yes

4. Is the manuscript presented in an intelligible fashion and written in standard English?

Reviewer #1: Yes

Reviewer #2: Yes

Reviewer #3: Yes

Reviewer #4: Yes

Reviewer #5: Yes

Reviewer #6: Yes

Reviewer #7: Yes

5. Review Comments to the Author

Reviewer #1: All in all a sound report discussing utility of the Pocket Smell Test vs. symptom-based screening for SARS-CoV-2. Not an exceptionally innovative or surprising study, but sound, practical, and acceptable pending revisions and statistical review.

Limitations:

-Acknowledged power analysis, but still a single-institution, smaller study, ethnically homogenous population. And only adults as well? Pregnant vs. non-pregnant? Needs to be expanded to larger, more diverse demographics in further work, and limitations should be mentioned.

-Another limitation related to time. The study took place between October 2020 and February 2021. The results may not still apply to a more vaccinated population and/or amidst quite different SARS-CoV-2 strains.

General:

-Why not focus results (in the Abstract and Table 2) moreso on the 2-odor test? Higher sensitivity, lower false negative rate, slightly not significantly higher AUC. Seems a better screening test than 8-odor? The 8-odor test is ~50% sensitive, better than symptom-based screening but far from ideal amidst the pandemic.

-It would be quite interesting to compare PSTs to home COVID-19 tests at this stage of the pandemic. This seems worth mentioning, from a cost aspect as well.

-Given age differences in hyposmia vs. normosmia populations, were the PSTs equally sensitive and specific in both groups, or more false positives in older populations and false negatives in younger populations? This seems worth mentioning if used as a screening tool.

Abstract:

-"Twenty-eight had a SARS-CoV-2 positive patients (31.8%) and 52 SARS-CoV-2 negative patients" miswording?

-"Smell testing is superior to symptom screening for identifying SARS-CoV-2 infection." Perhaps clarifying "In this study" given limitations as above

Introduction:

-Line 84: ambulatory; this study was ED/hospital-based

Methods:

-Line 96: enrollment (line 96)

-Line 133 : out of N=308

-Lines 135, 147: chi-squared

-148: p-values

Results:

-Line 155: sample or populations?

-Table 1: fine for "ED in maximum recorded temperature" since defined below table

Discussion:

-Line 242: et al.

-Line: 243: three day; three-day or every three day?

-Line 246: Cochrane

Reviewer #2: Thank you for giving me the opportunity to review this interesting piece of work which can be implicated in public health. With problems in detecting COVID19 and its diagnosis, such work may present promising directions for public health.

Reading through the abstract gives the reader the impression that this is a half-baked work while after reading the paper the truth is that the abstract falls short of fully capturing and introducing your work. I strongly recommend rewriting the whole abstract to reflect the true level of your work. Simple edits or adding few statements will not do the work.

The methods are written in thorough details.

Table 1: since you are not doing any form of randomization, I think there is no need to compare both groups statistically. This fallacy creates the impression that they were randomized which is not true for your study. I recommend removing the p-values as their interpretation is problematic.

Table 2: the table has so much information that makes it very difficult to read and follow through. I strongly recommend separating the table into two tables.

It is better that p-values to be reported without leading zeros.

I also suggest introducing the benefits/ implications of this test (convenient, accessible, cheap) in the introduction and putting it within the context of public health responses.

Reviewer #3: *There are some ambiguous sentences in the abstract section. For example: “…Twenty-eight had a SARS-CoV-2 positive 49 patients (31.8%) …” And the abstract should be clearer, and if possible, fascinating with parsimony.

* Eligibility criteria are not clearly elucidated; inclusion and exclusion criteria should be detailed. This is particularly important as the study focused in patients visiting the emergency department.

* In the limitation section/discussion part, some points should be raised, including: use of this olfaction-based screening should be deferred in those with reported baseline smell problems or smell loss (diminution).

Reviewer #4: This diagnostic accuracy study recruited 304 COVID suspects for comparing the sensitivities and specificities of symptom screening and smell test against test positivity. It then reported that the sensitivities of 8 – odor PST, 2 – odor test and symptom screen were 53.4%, 72.7% and 31.8% respectively; the specificities were 83.8%, 74.5% and 75.9%, respectively. The AUC for PST was 0.68, which improved to 0.79 for multiple variables. It then concluded that smell testing, particularly the two – item smell test could be a simple, affordable screening tool.

It must be noted that the convenience sample of patients is really made up of two distinct groups: one group consisted of COVID suspects while the second group consisted of patients for admission who may or may not be COVID suspects. While there was no separate descriptions of the symptoms and COVID test results of these two groups of patients, I think that the second group is the group that clinically reflects the general population for COVID screening, wherein the pretest probability of turning positive is lower, compared to the first group.. This may be the reason why the authors conducted subgroup analyses for the 8 – odor PSTs of asymptomatic and unexposed patients. Their AUCs were comparable with the 8 – odor PST for the entire sample which probably meant that the two groups of sampled patients could really be treated as one, although this was not pointed out.

I note that only 2.3% of the 216 COVID negative patients and 10% of the 88 COVID positive patients reported anosmia or ageusia. This does not appear to be consistent with other studies which reported anosmia as the most common symptom. I think this merits some explanation.

I think the authors should explain why the 2 odor test had a higher sensitivity but lower specificity compared to the 8 odor test because this is counter intuitive.

The results section in the abstract is also difficult to understand for me, the third sentence particularly. I think they should stick to the key findings in the abstract.

The authors cut short the study when the prevalence of COVID went up to 25% positivity. How does this affect the applicability of the study in a non-surge setting? Should sensitivity analyses be done?

I also did not note if this research was approved by the ethics committee of their institution.

Reviewer #5: The study is interesting and well conducted expecialyy in the methods. The authors conclude that smell testing was able to identify SARS-CoV-2 infection with a 73% of sensitivity and 75% od specifity even if was not investigated on asymptomatic subjects. However Bianco et al showed that alteration of sense of smell represent a prodromal symptom of infection, they coul add this reference as follow:Bianco, Maria Rita, et al. "Alteration of smell and taste in asymptomatic and symptomatic COVID-19 patients in Sicily, Italy." Ear, Nose & Throat Journal 100.2_suppl (2021): 182S-185S.

Reviewer #6: This study is apt and timely. It stands to contribute significantly to existing body of knowledge. However, the following revisions should be considered;

Title: The may benefit from inclusion of location of the study.

Abstract: The background to the abstract is somewhat over summarized with the objective(s) of the study not stated .

Introduction: The authors should consider differentiating asymptomatic from presymptomatic state of SARS-CoV-2 infections.

Methods:

1.The exclusion criteria used in this study should be stated clearly

2. It is unclear the measures used to mitigate against testing any participant more than once.

3. The authors also need to bring out clearly how the smell test was conducted, were the research clinicians trained on

conducting the smell test?

4. It is unclear if the those with anosmia/smell anomalies from any other conditions were identified and excluded from the study.

5.It appears a test of difference of means (unpaired student t test) was used for the comparison of age as well as temperature between the two categories but the authors presented it as if it was chi square.

6.It appears the age of the study participants in the two groups are skewed, it is imperative that the test of assessment of normality used should be stated otherwise an appropriate non parametric test should be used and stated.

Results: The two additional ROC model used " hyposmia & a 2-odor model" were not mentioned in the methods, it is important that mention is made of it and the rationale for it provided in the methods before stating such in the results.

Additional comment: It is important that the authors bring to light the limitation(s) associated with convenience sampling used particularly with regards to generalization of the findings

Reviewer #7: This is a well executed study and well written paper that examines a question many front line clinicians have asked themselves: how predictive is loss of smell of a diagnosis of COVID-19?

The authors assessed smell using a validated clinically relevant tool.

Couple of suggestions

-would suggest including in table 1 the smell test results frequencies (each one and aggregate).

-Also in table 1 would include variables that are significant in table s1.

-what is the difference between history and symptom of loss of smell, the latter presumably without formal testing? The numbers appear small for both. What happens when both are combined? How good is a model of history and/or symptom of loss of smell/taste plus 2 smell test (or are numbers too small)? How do authors explain the discrepancy between self reported loss of smell and smell test results?

-Table 2 is difficult to follow and should be perhaps split into two separate table or presented in a simplified table?

-a secondary multivariate analysis (logistic regression of loss of smell, perhaps stratified on COVID-19 status?) that includes variables with significant differences in bivariate analysis from table 1 and table s1 could help identify confounders that one has to think about when using this approach.

6. PLOS authors have the option to publish the peer review history of their article (what does this mean?). If published, this will include your full peer review and any attached files.

Reviewer #1: No

Reviewer #2: No

Reviewer #3: **Yes: **Subah Abderehim Yesuf

Reviewer #4: **Yes: **Jose Acuin

Reviewer #5: **Yes: **Eugenia Allegra

Reviewer #6: **Yes: **Tolulope Olumide Afolaranmi

Reviewer #7: No

---

## [Author Response · Author response to Decision Letter 0]

14 Feb 2022

February 14, 2022

Dear Editor and Reviewers,

Thank you for your many helpful comments and questions regarding our manuscript titled Performance of formal smell testing and symptom screening for identifying SARS-CoV-2 infection.

We have provided our responses following your comments. When appropriate we indicate the line numbers where we revised the manuscript.

In response to a concern from Reviewer #6 regarding participants being smell tested more than once (multiple visits to the ED), we reviewed our non-deidentified data set and found nine instances where a patient participated a second time. We excluded those second visits from our analytic data set, reran the analyses, and updated the tables and figures. Results were minimally (non-significantly) changed.

Regards,

James W Keck, MD MPH

Assistant Professor

Department of Family and Community Medicine

University of Kentucky

Comments to the Author Authors’ Response

Reviewer 1 

Acknowledged power analysis, but still a single-institution, smaller study, ethnically homogenous population. And only adults as well? Pregnant vs. non-pregnant? Needs to be expanded to larger, more diverse demographics in further work, and limitations should be mentioned. 

Authors’ Response

This limitation was acknowledged (lines 273-4). As mentioned in the methods (line 88) this study was limited to adults. Pregnant women were eligible, but we did not enroll any. As shown in table 1, 17% (49/295) participants were non-white and 3% (10/295) were Hispanic.

Another limitation related to time. The study took place between October 2020 and February 2021. The results may not still apply to a more vaccinated population and/or amidst quite different SARS-CoV-2 strains.

Authors’ Response

Added additional limitation statement: “Fourth, the epidemiologic context of the pandemic during our study period (e.g., predominant circulating virus variants and vaccine coverage) may influence study results and impact their generalizability to other pandemic contexts.” (lines 285-288)

Why not focus results (in the Abstract and Table 2) moreso on the 2-odor test? Higher sensitivity, lower false negative rate, slightly not significantly higher AUC. Seems a better screening test than 8-odor? The 8-odor test is ~50% sensitive, better than symptom-based screening but far from ideal amidst the pandemic. 

Authors’ Response

We adhered to our study protocol for the analysis which was designed to assess the 8-odor pocket smell test as a diagnostic/screening tool for SARS-CoV-2 infection. As defined in the protocol we individually assessed the discriminatory performance of each of the eight odors and reported those results including a subgroup analysis looking at the two odors with best performance.

It would be quite interesting to compare PSTs to home COVID-19 tests at this stage of the pandemic. This seems worth mentioning, from a cost aspect as well. 

Authors’ Response

We agree and the manuscript you reviewed included a comparison of smell testing to the performance of antigen based tests (lines 255-8).

Given age differences in hyposmia vs. normosmia populations, were the PSTs equally sensitive and specific in both groups, or more false positives in older populations and false negatives in younger populations? This seems worth mentioning if used as a screening tool. 

Authors’ Response

Although the age distribution of participants with hyposmia was on average 5 years older than those with Normosmia, the PST discriminated SARS-CoV-2 infection equally well across age groups. We ran two sub-analyses that split our sample into younger/older age groups (50th percentile and 75th percentile) and generated ROC models. The AUC was quite similar across groups: 50th percentile 0.71 vs 0.75 (older group) and 75th percentile 0.71 vs 0.77 (older group). 95% CIs were overlapping. We mention in the discussion that the robust performance of the PST across age groups (lines 236-239).

Abstract:

-"Twenty-eight had a SARS-CoV-2 positive patients (31.8%) and 52 SARS-CoV-2 negative patients" miswording? 

Authors’ Response: Corrected to read “Twenty-eight of the SARS-CoV-2 positive patients (31.8%)”

-"Smell testing is superior to symptom screening for identifying SARS-CoV-2 infection." Perhaps clarifying "In this study" given limitations as above 

Authors’ Response: Added “in our study.”

Introduction:

-Line 84: ambulatory; this study was ED/hospital-based 

Authors’ Response: Health care (at least in the USA) received in an emergency room is considered ambulatory care – see this WebMD page as a reference: https://www.webmd.com/health-insurance/terms/ambulatory-patient-services

Methods:

-Line 96: enrollment (line 96)

-Line 133: out of N=308

-Lines 135, 147: chi-squared

-148: p-values 

Authors’ Response: Corrections made.

Results:

-Line 155: sample or populations?

-Table 1: fine for "ED in maximum recorded temperature" since defined below table 

Authors’ Response: Changed “sample” to “participants. Changed “emergency department” to ED in Table 1.

Discussion:

-Line 242: et al.

-Line: 243: three day; three-day or every three day?

-Line 246: Cochrane 

Authors’ Response: Corrections made. Changed to “third day”

Reviewer #2 

Reading through the abstract gives the reader the impression that this is a half-baked work while after reading the paper the truth is that the abstract falls short of fully capturing and introducing your work. I strongly recommend rewriting the whole abstract to reflect the true level of your work. Simple edits or adding few statements will not do the work. 

Authors’ Response: There were several typos in the results section of the abstract. These have been corrected which should improve the readability of the abstract.

Table 1: since you are not doing any form of randomization, I think there is no need to compare both groups statistically. This fallacy creates the impression that they were randomized which is not true for your study. I recommend removing the p-values as their interpretation is problematic. 

Authors’ Response

Because of the study design (assessment of diagnostic tool) we could not use randomization to minimize the risk of confounders. Hence it is important to describe and compare the attributes of the two groups (Sars-CoV-2 positive and negative) to identify potential confounders that would need to be controlled for in the analysis.

Table 2: the table has so much information that makes it very difficult to read and follow through. I strongly recommend separating the table into two tables. 

Authors’ Response

To simplify the table we removed the sensitivity and specificity rows as those values are listed in the text. We also removed the “2-odor” screening test because this was a sub-analysis of the smell testing performance. We believe the various performance metric of the screening tests are best compared side-by-side in a single table. 

It is better that p-values to be reported without leading zeros. 

Authors’ Response

PLOS One journal publishes P-values with leading zeros.

I also suggest introducing the benefits/ implications of this test (convenient, accessible, cheap) in the introduction and putting it within the context of public health responses. We mention these benefits in the discussion (lines 241-44). 

Authors’ Response

As the focus of the study was evaluating smell testing discriminatory performance (and not cost or accessibility) we did not discuss these potential attributes in the introduction.

Reviewer #3 

There are some ambiguous sentences in the abstract section. For example: “…Twenty-eight had a SARS-CoV-2 positive 49 patients (31.8%) …” And the abstract should be clearer, and if possible, fascinating with parsimony. 

Authors’ Response

There were several typos in the results section of the abstract. These have been corrected which should improve the readability of the abstract.

Eligibility criteria are not clearly elucidated; inclusion and exclusion criteria should be detailed. This is particularly important as the study focused in patients visiting the emergency department. 

Authors’ Response

Because we wanted a diverse participant sample, we had few inclusion criteria. Those criteria are listed in the Methods-study population section and included 1) adults 2) seeking care at the study hospital with anticipated SARS-CoV-2 testing due to either 3) self-reported COVID-19 symptom on screening or close contact with a known case or 4) planned hospital admission. 

In the limitation section/discussion part, some points should be raised, including: use of this olfaction-based screening should be deferred in those with reported baseline smell problems or smell loss (diminution). 

Authors’ Response

Given the previously published inconsistency in self-reported versus measured loss/alteration of smell (see intro and references 5 and 8) and a similar finding in our study, we do not believe excluding individuals with self-reported smell problems is useful. To confirm this, we conducted a sub-analysis (n=267) restricted to participants reporting NO history of smell problems. The AUC for the ROC model was 0.72 – slightly less (but not statistically different) than the AUC for the entire sample which included people reporting a history of smell problems. This suggests the 8-odor smell testing screening approach performs similarly when including participants with a reported history of smell problems.

Reviewer #4 

It must be noted that the convenience sample of patients is really made up of two distinct groups: one group consisted of COVID suspects while the second group consisted of patients for admission who may or may not be COVID suspects. While there was no separate descriptions of the symptoms and COVID test results of these two groups of patients, I think that the second group is the group that clinically reflects the general population for COVID screening, wherein the pretest probability of turning positive is lower, compared to the first group. This may be the reason why the authors conducted subgroup analyses for the 8 – odor PSTs of asymptomatic and unexposed patients. Their AUCs were comparable with the 8 – odor PST for the entire sample which probably meant that the two groups of sampled patients could really be treated as one, although this was not pointed out. 

Authors’ Response

We agree with your comment. We wanted to include asymptomatic patients (not only COVID-19 suspects by symptom screening) to assess the performance of smell testing in both groups. As you point out, we conducted a sub-group analysis of the asymptomatic patients and found that smell testing performed similarly in this group.

I note that only 2.3% of the 216 COVID negative patients and 10% of the 88 COVID positive patients reported anosmia or ageusia. This does not appear to be consistent with other studies which reported anosmia as the most common symptom. I think this merits some explanation. 

Authors’ Response

A limitation of symptom-based screening (like self-reported alteration in smell or taste) is that it misses the 50% or more of people infected with SARS-CoV-2 that are either pre-symptomatic at the time point of screening or have an asymptomatic infection. Since we smell tested and PCR-tested asymptomatic patients, a smaller proportion of our sample reported altered smell or taste as compared to the studies we mention in the discussion (lines 248-257) that suffered from substantial selection and measurement biases.

I think the authors should explain why the 2 odor test had a higher sensitivity but lower specificity compared to the 8 odor test because this is counter intuitive 

Authors’ Response

As indicated in Table 2, for calculating the sensitivity and specificity of the 8-odor model we used the NHANES definition of hyposmia (misidentifying 3 or more odors) as our cut point. For the 2-odor model, we used misidentification of 1 odor as the cut point. This choice of cut points affected the calculated sensitivity and specificity of the smell tests. If for instance we defined abnormal smell testing performance on the 8-odor test as misidentifying one or more odors, the test would be very sensitive but not specific for SARS-CoV-2 infection.

The results section in the abstract is also difficult to understand for me, the third sentence particularly. I think they should stick to the key findings in the abstract. 

Authors’ Response

There were several typos in the results section of the abstract. These have been corrected which should improve the readability of the abstract.

The authors cut short the study when the prevalence of COVID went up to 25% positivity. How does this affect the applicability of the study in a non-surge setting? Should sensitivity analyses be done? 

Authors’ Response

The discriminatory performance of a screening or diagnostic test (sensitivity and specificity) is independent of disease prevalence. We included in Table 2 three levels of disease prevalence to show how the positive and negative predictive values of the different screening approaches change depending on disease prevalence.

I also did not note if this research was approved by the ethics committee of their institution. 

Authors’ Response

We received Institutional Review Board (equivalent of ethics committee) approval as noted in lines 94-96.

Reviewer #5 

The study is interesting and well conducted expecialy in the methods. The authors conclude that smell testing was able to identify SARS-CoV-2 infection with a 73% of sensitivity and 75% od specifity even if was not investigated on asymptomatic subjects. However Bianco et al showed that alteration of sense of smell represent a prodromal symptom of infection, they coul add this reference as follow: Bianco, Maria Rita, et al. "Alteration of smell and taste in asymptomatic and symptomatic COVID-19 patients in Sicily, Italy." Ear, Nose & Throat Journal 100.2_suppl (2021): 182S-185S. 

Authors’ Response

We, in fact, investigated smell testing in asymptomatic patients. As described in Table 1, many of our participants had no symptoms (were asymptomatic or prodromal). We agree that altered smell can occur in presymptomatic (prodomal) patients, which is why we conducted formal smell testing and found it more sensitive than reported symptoms in identifying SARS-CoV-2 infection.

Reviewer #6 

Title: The may benefit from inclusion of location of the study. 

Authors’ Response

We felt that including a location in the title might distract readers as the objective of the study was to evaluate the performance of a SARS-CoV-2 screening test (smell testing) and not describe a location-based phenomenon

Abstract: The background to the abstract is somewhat over summarized with the objective(s) of the study not stated 

Authors’ Response

We added a statement in the abstract background about the objective: We evaluated the performance of formal smell testing to identify SARS-CoV-2 infection. (lines 35-36)

Introduction: The authors should consider differentiating asymptomatic from presymptomatic state of SARS-CoV-2 infections. 

Authors’ Response

We are unclear what this reviewer means by differentiating the asymptomatic and presymptomatic state of infection. We believe these are commonly used terms that the readers of PLOS One will understand.

Methods:

1.The exclusion criteria used in this study should be stated clearly

2. It is unclear the measures used to mitigate against testing any participant more than once.

3. The authors also need to bring out clearly how the smell test was conducted, were the research clinicians trained on

conducting the smell test?

4. It is unclear if the those with anosmia/smell anomalies from any other conditions were identified and excluded from the study.

5.It appears a test of difference of means (unpaired student t test) was used for the comparison of age as well as temperature between the two categories but the authors presented it as if it was chi square.

6.It appears the age of the study participants in the two groups are skewed, it is imperative that the test of assessment of normality used should be stated otherwise an appropriate non parametric test should be used and stated. 

Authors’ Responses

1. Added an exclusion criteria sentence “We excluded patients with an altered level of consciousness and minors.” (line 94)

2. Thank you for bringing this to our attention. We reviewed the source data and using the unique medical record number of each patient-participant identified 9 instances of repeat encounters with smell tasting data. We kept the data from the first of the two encounters and re-ran the data analysis with the 295 unique participants. There were small, statistically insignificant changes across most of our descriptive measures. All results in the manuscript and supporting material have been updated using the de-duplicated data set.

3. As noted in the methods section (lines 113-115) the PST smell test is designed to be self-administered. Participants scratch and odor strip and circle one of four multiple choice options to identify the odor.

4. We asked participants about a history of loss of smell (not recent) and reported those results in supplemental table S1. 28 participants reported a history of loss of smell. We did not exclude these patients from our primary analysis. 

5. The descriptive statistics of age and temperature appearing in table 1 were generated with t tests. For measures of association of the linear predictor variables with the binary outcome variable (SARS-CoV-2 infection) we used logistic regression. The methods section (lines136-7) was updated to reflect this.

6. Thank you for bringing this to our attention. We inspected the age distribution visually and statistically and it was not normally distributed. We used the Wilcoxon rank-sum test to assess difference in age means between samples and updated the methods and results to reflect this analysis. 

Results: The two additional ROC model used " hyposmia & a 2-odor model" were not mentioned in the methods, it is important that mention is made of it and the rationale for it provided in the methods before stating such in the results. 

Authors’ Response

We added substantial additional detail to the methods section to describe the six ROC models presented in table 3 (lines 143-153)

Additional comment: It is important that the authors bring to light the limitation(s) associated with convenience sampling used particularly with regards to generalization of the findings 

Authors’ Response

We recognize this limitation and mentioned it first in our paragraph describing study limitations (lines 273-275), stating “we recruited a convenience sample of patients seeking care at the emergency department, and we cannot generalize the findings of our study to other populations, such as asymptomatic people in the general population.”

Reviewer #7 

would suggest including in table 1 the smell test results frequencies (each one and aggregate). Also in table 1 would include variables that are significant in table s1. 

Authors’ Response

We report the smell test result frequencies in figure 1 as point estimates with 95% CI bars by SARS-CoV-2 test status. We believe reporting the same data in tabular form would be redundant.

what is the difference between history and symptom of loss of smell, the latter presumably without formal testing? The numbers appear small for both. What happens when both are combined? 

Authors’ Response

The symptom “loss of taste and/or smell” was part of the emergency department’s COVID-19 screening questionnaire as described in the methods section (lines 90-92). The “history of loss of smell” was a question on our patient survey (methods section lines 112-113) about pre-existing health conditions and was intended to identify patients with underlying olfactory disorder. Since one question asks about an acute symptom and the other about a pre-existing condition, combining the responses would not be appropriate.

How good is a model of history and/or symptom of loss of smell/taste plus 2 smell test (or are numbers too small)? 

Authors’ Response

Using the three classifiers 1) soap odor, 2) smoke odor, and 3) reported loss of taste and/or smell, a ROC model has an AUC of 0.76 (295 observations) – which is essentially the same as the 2-odor ROC model (AUC=0.75).

How do authors explain the discrepancy between self reported loss of smell and smell test results? 

Authors’ Response

We addressed this in the introduction – lines 72-77. In short, people are often not aware that they have a diminished sense of smell, hence the interest in formal smell testing as a SARS-CoV-2 screening tool.

Table 2 is difficult to follow and should be perhaps split into two separate table or presented in a simplified table? 

Authors’ Response

To simplify the table we removed the sensitivity and specificity rows as those values are listed in the text. We also removed the “2-odor” screening test because this was a sub-analysis of the smell testing performance. We believe the various performance metric of the screening tests are best compared side-by-side in a single table. 

a secondary multivariate analysis (logistic regression of loss of smell, perhaps stratified on COVID-19 status?) that includes variables with significant differences in bivariate analysis from table 1 and table s1 could help identify confounders that one has to think about when using this approach. 

Authors’ Response

We reported an “adjusted” ROC model that adjusted for age, gender, corticosteroid nasal spray use, measured fever, and reported cough as described in methods (lines 149-150) and in results (lines 216-218). We adjusted for age and nasal spray use because these were associated with smell test performance, and we adjusted for fever and cough because these were associated with SARS-CoV-2 infection (as reported in table 1 and S1.

---

## [Decision Letter · Decision Letter 1]

1 Mar 2022

PONE-D-21-38950R1Performance of formal smell testing and symptom screening for identifying SARS-CoV-2 infectionPLOS ONE

Dear Dr. Keck,

Thank you for submitting your manuscript to PLOS ONE. After careful consideration, we feel that it has merit but does not fully meet PLOS ONE’s publication criteria as it currently stands. Therefore, we invite you to submit a revised version of the manuscript that addresses the points raised during the review process.

We look forward to receiving your revised manuscript.

Kind regards,

Muhammad Tarek Abdel Ghafar, M.D

Academic Editor

PLOS ONE

Journal Requirements:

Reviewers' comments:

Reviewer's Responses to Questions

**Comments to the Author**

1. If the authors have adequately addressed your comments raised in a previous round of review and you feel that this manuscript is now acceptable for publication, you may indicate that here to bypass the “Comments to the Author” section, enter your conflict of interest statement in the “Confidential to Editor” section, and submit your "Accept" recommendation.

Reviewer #1: All comments have been addressed

Reviewer #3: All comments have been addressed

Reviewer #6: (No Response)

Reviewer #7: All comments have been addressed

2. Is the manuscript technically sound, and do the data support the conclusions?

Reviewer #1: Yes

Reviewer #3: Yes

Reviewer #6: Yes

Reviewer #7: Yes

3. Has the statistical analysis been performed appropriately and rigorously? 

Reviewer #1: I Don't Know

Reviewer #3: Yes

Reviewer #6: Yes

Reviewer #7: Yes

4. Have the authors made all data underlying the findings in their manuscript fully available?

Reviewer #1: Yes

Reviewer #3: (No Response)

Reviewer #6: Yes

Reviewer #7: Yes

5. Is the manuscript presented in an intelligible fashion and written in standard English?

Reviewer #1: Yes

Reviewer #3: Yes

Reviewer #6: Yes

Reviewer #7: Yes

6. Review Comments to the Author

Reviewer #1: Nice revisions; probably wouldn't advise referencing WebMD on reviewer responses, but not a barrier to acceptance. Acceptable pending PLOS One statistical review protocols and consideration of below points.

Line 125: Figure 1 -- caption meant to be here in the methods section?

Lines 131-132: understood but not quite grammatically clear

Lines 152: understood perhaps specifying what "best performance" means

Line 234: interesting perhaps unexpected given that nasal steroids did impact performance, correct? And assuming most people with rhinitis on nasal steroids?

Line 276: or negative PCR and sustained olfactory dysfunction that would decrease the already high specificity, yes? but no need to specifically note to avoid wordage.

Reviewer #3: * The authors have tried to absorb the important comments provided by the previous reviewers. The authors have digested the manuscript to an appreciable extent. As it stands, the manuscript is well written, and it should be considered for publication provided after inculcating the minors both from the reviewers and academic editor(s).

* The abstract presents an accurate synopsis of the paper.

*Discussion: There is some ambiguity in line (≠250 and ≠251): …the utility of this using this as..?

Reviewer #6: The authors have addressed most of the review comments. However, the quality of this manuscript will be improved if more information is added to the background to the abstract as well as the conclusion of the abstract.

Reviewer #7: Substantive issues were addressed

7. PLOS authors have the option to publish the peer review history of their article (what does this mean?). If published, this will include your full peer review and any attached files.

Reviewer #1: No

Reviewer #3: No

Reviewer #6: **Yes: **Tolulope Olumide Afolaranmi

Reviewer #7: No

---

## [Author Response · Author response to Decision Letter 1]

16 Mar 2022

Reviewer #1 

Line 125: Figure 1 -- caption meant to be here in the methods section? 

+ Yes – per PLOS One author guidelines the figure caption should be placed after the paragraph where it is first reference. Figure 1 is referenced in the preceding paragraph.

Lines 131-132: understood but not quite grammatically clear 

+ Removed clause “based on unique medical record number” to improve sentence syntax.

Lines 152: understood perhaps specifying what "best performance" means 

+ Clarified that the performance metric we used was AUC (area under the curve) 

Line 234: interesting perhaps unexpected given that nasal steroids did impact performance, correct? And assuming most people with rhinitis on nasal steroids? 

+ Looking at S1 Table, about 50% of sample reported seasonal allergies (believable in Kentucky). However, only 12% reported nasal steroid use. We modified this sentence (lines 234-36) to represent our analysis more accurately, which found that smell testing performance was similar between patients with/without sinusitis and seasonal allergies. We didn’t measure whether either condition was an effect modifier of the discriminatory ability of smell testing for SARS-CoV-2 infection. 

Line 276: or negative PCR and sustained olfactory dysfunction that would decrease the already high specificity, yes? but no need to specifically note to avoid wordage. 

+ We agree that this scenario is also possible and would decrease the specificity of the smell test.

Reviewer #3 

Discussion: There is some ambiguity in line (≠250 and ≠251): …the utility of this using this as..? 

+ We revised the sentence to read “The performance of smell testing in asymptomatic patients suggests its utility as a SARS-CoV-2 screening tool in asymptomatic populations, like employees at congregate work settings” for clarity.

Reviewer #6 

The quality of this manuscript will be improved if more information is added to the background to the abstract as well as the conclusion of the abstract. 

+ We minimally revised the background section of the abstract for improved clarity. Respectfully, we are unclear on what additional information would be helpful in the abstract. An abstract provides a concise description of the study and invites the reader to review the full manuscript for greater detail. As PLOS One is open access, anyone can read the manuscript in its entirety.

---

## [Decision Letter · Decision Letter 2]

30 Mar 2022

Performance of formal smell testing and symptom screening for identifying SARS-CoV-2 infection

PONE-D-21-38950R2

Dear Dr. Keck,

We’re pleased to inform you that your manuscript has been judged scientifically suitable for publication and will be formally accepted for publication once it meets all outstanding technical requirements.

Kind regards,

Muhammad Tarek Abdel Ghafar, M.D

Academic Editor

PLOS ONE

Additional Editor Comments (optional):

Reviewers' comments:

Reviewer's Responses to Questions

**Comments to the Author**

1. If the authors have adequately addressed your comments raised in a previous round of review and you feel that this manuscript is now acceptable for publication, you may indicate that here to bypass the “Comments to the Author” section, enter your conflict of interest statement in the “Confidential to Editor” section, and submit your "Accept" recommendation.

Reviewer #1: All comments have been addressed

2. Is the manuscript technically sound, and do the data support the conclusions?

Reviewer #1: Yes

3. Has the statistical analysis been performed appropriately and rigorously? 

Reviewer #1: I Don't Know

4. Have the authors made all data underlying the findings in their manuscript fully available?

Reviewer #1: Yes

5. Is the manuscript presented in an intelligible fashion and written in standard English?

Reviewer #1: Yes

6. Review Comments to the Author

Reviewer #1: (No Response)

7. PLOS authors have the option to publish the peer review history of their article (what does this mean?). If published, this will include your full peer review and any attached files.

Reviewer #1: No

---

## [Editor Report · Acceptance letter]

4 Apr 2022

PONE-D-21-38950R2 

Performance of formal smell testing and symptom screening for identifying SARS-CoV-2 infection 

Dear Dr. Keck:

I'm pleased to inform you that your manuscript has been deemed suitable for publication in PLOS ONE. Congratulations! Your manuscript is now with our production department. 

Kind regards, 

on behalf of

Prof Muhammad Tarek Abdel Ghafar 

Academic Editor

PLOS ONE